# A Debate on Surgical and Nonsurgical Approaches for Obstructive Sleep Apnea: A Comprehensive Review

**DOI:** 10.3390/jpm13091288

**Published:** 2023-08-23

**Authors:** Andreea Zabara-Antal, Ionela Grosu-Creanga, Mihai Lucian Zabara, Andrei Tudor Cernomaz, Bogdan Mihnea Ciuntu, Oana Melinte, Cristian Lupascu, Antigona Carmen Trofor

**Affiliations:** 1Clinical Hospital of Pulmonary Diseases Iași, 700116 Iasi, Romania; andreeazabara@yahoo.com (A.Z.-A.); ionela.grossu@yahoo.com (I.G.-C.); rohozneanuoana@yahoo.com (O.M.); atrofor@yahoo.com (A.C.T.); 2Pulmonary Department, Faculty of General Medicine, University of Medicine and Pharmacy “Grigore T. Popa”, 700115 Iasi, Romania; 3Doctoral School, Faculty of Medicine, University of Medicine and Pharmacy “Grigore T. Popa”, 700115 Iasi, Romania; 4Department of Surgery, Faculty of Medicine, University of Medicine and Pharmacy “Grigore T. Popa”, 700115 Iasi, Romania; mihai-lucian.zabara@umfiasi.ro (M.L.Z.); bogdan-mihnea.ciuntu@umfiasi.ro (B.M.C.); cristian.lupascu@umfiasi.ro (C.L.); 5Clinic of Surgery (II), St. Spiridon Emergency Hospital, 700111 Iasi, Romania; 6Regional Institute of Oncology, 700483 Iasi, Romania

**Keywords:** sleep apnea, obstructive sleep apnea, obesity, obesity hypoventilation syndrome, CPAP, BiPAP, bariatric surgery

## Abstract

Regular and unobstructed breathing during the night is the prerequisite for an undisturbed and restful sleep. The most prevalent nocturnal breathing disturbance with morbid consequences is obstructive sleep apnea syndrome. The prevalence of obstructive sleep apnea (OSA) is increasing, and a significant number of patients with OSA are undiagnosed. On the other hand, the “obesity epidemic” is a growing concern globally. A sleep test is required to diagnose sleep apnea and to individualize therapy. A multidisciplinary approach is the key to success. This narrative review presents a debate on whether surgery is a friend or a foe in the treatment of sleep apnea. Depending on the type and severity of the apnea, the causal factor, and the presence of obesity and hypercapnia as well as the associated pathologies, the optimal therapeutic method is determined for each individual case. The article concludes that each case is unique, and there is no ideal method. Positive pressure ventilation, although a therapeutic gold standard, has its disadvantages extensively discussed in this paper. Nevertheless, it is necessary prior to any surgical intervention, either for the curative treatment of the causal factor of apnea or for elective surgery for another condition. The anesthetic risks associated with the presence of apnea and obesity should not be neglected and should form the basis for decision making regarding surgical interventions for the treatment of sleep apnea.

## 1. Introduction

Sleep occupies a quarter of a person’s life, having a major impact on its quality.

This is a physiological and reversible process characterized by a decrease in sensitivity and response to external stimulus. There are over 90 sleep disorders that have a significant impact both at an individual level, resulting in reduced quality of life, and at a societal level, leading to decreased productivity and high medical costs. In the long term, they result in an overloading of the nervous system, with impactful effects on the entire body [1]. Regular and unobstructed breathing during the night is the prerequisite for an undisturbed and restful sleep. The most prevalent nocturnal breathing disturbance with morbid consequences is the obstructive sleep apnea syndrome [2]. The prevalence of obstructive sleep apnea (OSA) is increasing, and a significant number of patients with OSA remain undiagnosed [3]. Nowadays, roughly 1 billion cases are documented worldwide [4]. In a study conducted in 2016, Schaal et al. emphasized the idea that 24% of patients have an asymptomatic form [5]. Despite being current, it remains a controversial and an insufficiently debated topic. The type of patient, symptoms, and associated risks do not “sound the alarm” as is the case with other breathing-related conditions. Individuals with risk factors, although easily identifiable, are still not promptly signaled and directed to sleep laboratories by healthcare professionals, delaying specialized treatment. On the other hand, we should bear in mind that the “obesity epidemic” is a globally growing concern. Obesity is the most common contributing factor to sleep apnea [6].

Sleep apnea syndrome represents the restriction or cessation of airflow during sleep, either due to the collapse of the upper airways (obstructive phenomena) or the inhibition of the respiratory centers (central phenomena) [7].

The effects of sleep on respiration can be summarized as an increase in airway resistance, a decrease in the respiratory rate (by 0.5–1.5 breaths per minute), a decrease in the metabolic rate by 10–15%, reduction in chemosensitivity (20–50%), an increase in the partial pressure of carbon dioxide (PaCO2) by 2–8 mmHg, a decrease in partial pressure of oxygen (PaO2) by 3–10 mmHg, and a decrease in oxygen saturation.

## 2. Materials and Methods

We conducted a literature search of PubMed and MEDLINE for studies and different publications for theoretical information in order to evaluate and debate the therapeutic opportunities on the topic of sleep apnea. The terms used for searching were “sleep apnea”, “obesity”, “CPAP”, “BPAP”, and “bariatric surgery”. Observational studies, including prospective and retrospective cohort studies, meta-analyses, and case reports, were included.

## 3. Result

### 3.1. Multidisciplinarity—Who Do We Collaborate with?

When we refer to obstructive sleep apnea syndrome, we are also referring to comorbidities. A multidisciplinary approach is the key to management. Starting from hypoxia, the disease is associated with cardiologic, ophthalmologic [8], and metabolic impairments. A sleep study is required to diagnose sleep apnea and to individualize therapy [2].

Through the pathophysiological mechanism, sleep apnea leads to the onset or worsening of preexisting ophthalmologic conditions. Patients with obstructive sleep apnea syndrome (OSAS) may have a higher risk of developing glaucoma-related changes, which is why they should undergo regular ophthalmologic examinations to detect glaucomatous changes and possible disease progression. Sleep-related manifestations contribute to severe ocular surface and retinal disorders, manifested by a reduction in retinal nerve fiber layer thickness [8].

According to the latest literature, 30% of patients with obstructive sleep apnea syndrome (OSAS) develop acute coronary syndromes, 25% develop congestive heart failure, 50% develop systemic arterial hypertension, and 60% are at risk of suffering a stroke. Sympathetic arousal, endothelial dysfunction, inflammation, oxidative stress, and metabolic dysfunction caused by hypoxia are the underlying factors in the development of cardiovascular diseases. The characteristic element of hypoxemia in OSAS is the recurrence of desaturations and subsequent resaturations, which play a crucial role in certain molecular mechanisms of cardiovascular diseases. An increase in the average diastolic blood pressure over 24 h due to the nondipper pattern and a daytime increase in systemic arterial hypertension due to repeated airway obstructions during the night are distinctive features. In a study by Haentiens et al., conducted on 572 patients, the average blood pressure over 24 h significantly decreased after continuous positive airway pressure (CPAP) treatment [9].

In another study, 54 patients diagnosed with both ischemic heart disease and OSAS (25 treated with CPAP and 24 without CPAP) were followed over a 5-year period. Patients with similar characteristics, such as age, body mass index (BMI), smoking history, diabetes, dyslipidemia, severity of coronary artery disease, and left ventricular ejection fraction were selected. The study concluded that there is an increased cardiovascular risk in patients with untreated OSAS, including mortality, myocardial infarction, hospitalization for congestive heart failure, and revascularization [10].

Prior to any elective surgical intervention, individuals with risk factors for OSAS are referred to undergo a sleep test. Risk factors include obesity, changes in the oropharyngeal area, daytime symptoms (such as sleepiness and headaches), and nocturnal symptoms (snoring and fragmented sleep) [11]. If the diagnosis is confirmed, it is mandatory to undergo positive pressure ventilation for 4–6 weeks prior to surgery. During the preanesthetic consultation, the anesthesiologist assesses the risk factors and requests a sleep test. The patient visits an accredited sleep laboratory. The process is the same for bariatric surgeries or other elective surgical interventions. Surgical treatment will only be performed if there is evidence of adherence to treatment, tolerance of the ventilation device, and effectiveness of treatment. This information is obtained by reading the compliance card of the ventilation device. Only after that, the patient is suitable for anesthesia and can benefit from intervention. The prevalence of OSA implies significant risks for perioperative settings as patients are vulnerable to cardiopulmonary complications, critical care requirements and unexpected death [4].

### 3.2. The Patient Profile with Obstructive Sleep Apnea Syndrome (OSAS)

The patient profile with OSAS usually includes obesity, exertional dyspnea, and day and night symptoms. If there is extreme obesity, signs of right heart failure and pulmonary hypertension may also be present. Restrictive ventilatory dysfunction occurs. Polycythemia and comorbidities are commonly associated. Obese patients have a higher prevalence of diagnoses such as heart failure, angina pectoris, and chronic pulmonary heart disease [12].

### 3.3. Diagnosis

A thorough medical history, accompanied by a complete clinical examination and the completion of specific questionnaires (Berlin, STOP-Bang, and Epworth), raises diagnostic suspicion. Further, a sleep study is performed. This can be conducted either in a sleep laboratory or as an ambulatory test.

### 3.4. Surgery: Friend or Foe?

Regarding the management of sleep apnea syndrome (OSAS), this paper aims to initiate a friendly debate. Can the advantages of surgical therapy outweigh the disadvantages in the case of OSAS?

Regarding sleep apnea patients, we should approach two directions. On the one hand, surgical treatment of OSA is important. On the other hand, the management of this condition in patients who present electively for another type of surgery should be improved. Any type of elective surgery should have the same management. Through the development of sleep laboratories and the performance of ambulatory tests, this is now possible [13].

Furthermore, we aim to describe therapeutic options and different approaches to this sleep disorder. The perioperative complications of OSA, the screening tools available for physicians to assess surgical patients, and the perioperative management of these patients are important tools that complete the management of the disease when it comes to a rather major solution such as surgery. However, there are some advantages and disadvantages of positive pressure ventilation and noninvasive ventilation that should also be taken into consideration and seem to favor, in some cases, the surgical approach.

First-line management of the OSAS involves sleep hygiene measures (sleeping in a well-ventilated and clean bedroom; avoiding the consumption of central nervous system stimulants such as coffee, alcohol, or “cola”-based soft drinks at least 4 h before bedtime; avoiding physical exercise within 2 h before bedtime; and avoiding using the bedroom as a workspace). Sleeping in a lateral position is helpful as the majority of respiratory events occur while sleeping in the supine position. In addition, elimination of causal factors should be addressed, such as weight loss or unresolved relevant ENT (ear, nose, and throat) pathology.

By increasing airway resistance due to edema and inflammation, smoking predisposes individuals to the development of sleep apnea, especially as 20% of patients with sleep apnea also have chronic obstructive pulmonary disease (COPD) [14]. In COPD caused by smoking (and also in sleep apnea related to smoking), cessation is an essential intervention and seems to be associated with barriers that can interfere with either the patient’s motivation to quit or motivating them to maintain the subsequent continuous abstinence when the condition is improving. Smoking cessation should be intensive and sustained, and more intensive reinforcing visits with healthcare professionals could maintain the motivation and increase the chance of continuous abstinence and adherence to a nonsmoking life [15]. As for nonsurgical therapeutic methods, we will refer to mandibular advancement devices, tongue advancement/stabilization devices, oral appliances, and positional therapy.

The gold-standard treatment remains positive pressure ventilation. Depending on the type and severity of apnea, options include CPAP (continuous positive airway pressure), APAP (automatic positive airway pressure), or BPAP (bilevel positive airway pressure) devices. They function as pneumatic splints, preventing respiratory collapse during sleep and restoring normal sleep architecture [16]. The reduction of attention deficits is particularly important for drivers, with patients reporting restful sleep from the first night of use.

### 3.5. The “Otorhinolaryngology Affair”

In 1991, Pringle and Croft introduced drug-induced sleep endoscopy, a technique invented for evaluating snoring and obstructive sleep apnea syndrome, as they observed the poor outcomes of uvulopalatopharyngoplasty (UPPP) in resolving OSAS. They observed that the obstruction is multilevel and dynamic, which is why a single surgery intervention cannot work for all patients. Depending on the site of obstruction, different surgical techniques are implied. Drug-induced sleep endoscopy (DISE) changes the operative indication in more than 40% of patients regarding the structures contributing to oropharyngeal and laryngeal obstruction. Moreover, polypectomy and nasal concha surgery should be performed if the site of obstruction is the nose, while adenoidectomy is recommended if the obstruction is considered to originate from the nasopharynx [15].

Tonsillectomy or uvulopalatopharyngoplasty are techniques used when addressing oropharyngeal obstruction. Surgery of the oropharynx includes lingual tonsillectomy, midline glossectomy with lingual plasty, glossopexy, limited mandibular osteotomies, genioglossus advancement, and maxillomandibular advancement. In extreme cases, a tracheostomy (method of bypassing the upper airways) is the only therapeutic intervention available [15].

Other therapeutic principles using oropharyngeal surgery include reducing critical occlusion pressure by calibrating the upper airways, increasing endolaryngeal pressure, and enhancing the activity of pharyngeal dilator muscles. An alternative to surgical treatment is a mandibular advancement device. It primarily addresses collapse at the base of the tongue, and its utility and titration can be determined through polysomnography while using the device or during drug-induced sleep endoscopy (DISE). However, it has adverse effects, such as difficulties in chewing, excessive salivation, dental and tongue discomfort, and dental and occlusal changes [17].

Tongue advancement/stabilization devices have similar results but with fewer adverse effects. However, they are not used very often.

Therapeutic devices for positional training are diverse and include a treatment effectiveness monitor. They assess sleep quality and its fragmentation, associating a decrease in sleep quality with the degree of snoring, and determine whether snoring and poor-quality sleep occur in the supine position [18].

### 3.6. Bariatric Surgery Pro and Cons

Regarding sleep apnea, when the patient is obese (or has obesity hypoventilation syndrome), most patients tend to opt for bariatric surgery. Prior to the intervention, the protocol requires a mandatory sleep test. Subsequently, patients with respiratory pauses during sleep will undergo treatment with positive ventilation using a BPAP (bilevel positive airway pressure) machine a few weeks prior to surgery. Afterwards, compliance, tolerance, and effectiveness of the therapy are assessed by reading the compliance stick of the device. The Epworth questionnaire provide information for reducing symptoms.

Following surgical intervention and weight loss, a repeat sleep study is conducted to determine the feasibility of continuing home ventilation with BPAP. We are referring to the use of BPAP even though the therapeutic approach in these cases is surgical. These are specific and important to mention in pre- and postoperative management interventions for patients of this category while always bearing in mind that there is a significant anesthetic risk that should be considered, addressed, and, if possible, lowered by these interventions. However, after significant weight loss, the diagnosis is sometimes no longer confirmed upon repeating the polysomnography.

Losing 10% of body weight leads to a 30% reduction in AHI (apnea–hypopnea index) [19]. On the other hand, there is a risk of rebound weight gain, in which case, with the reemergence of the risk factor, OSA (obstructive sleep apnea) reappears. In any case, a sleep test is necessary as well as a reattempt of treatment with the use of CPAP (continuous positive airway pressure) and reading the compliance card. Thus, even if surgical treatment is chosen, ventilation is still necessary for a certain period of time.

Of course, bariatric surgery has its advantages. A follow-up study (2009) showed how patients losing weight recovered sleep architecture, reduced AHI and symptoms, recovered normal blood pressure, and re-established quality of life [1]. Patients and healthcare professionals must constantly be reminded that positive pressure ventilation does not cure sleep apnea but rather treats the symptoms, restores normal sleep architecture, and eliminates respiratory events as long as it is used.

The disadvantages of using CPAP or BPAP home ventilation can be summarized as follows: psychological impact (often observed in younger patients), associated costs, negative effects on intimate life, and dependence on its continuous use. Some patients see this long-term use indication as a compromise and, eventually, a possible decrease in the quality of life. Of course, low tolerance because of ENT pathology is also important.

Regarding obesity surgery, there are, of course, limits and disadvantages. Morbid obesity, frequent in OSAS, has been associated with increased duration of mechanical ventilation and prolonged intensive care unit (ICU) hospitalization. On the other hand, prophylactic noninvasive ventilation (NIV) within the first 24 h after bariatric surgery leads to a significant decrease in ventilatory dysfunction and accelerates the recovery of pulmonary function. It has been proven that NIV in obese patients within the first 48 h after extubation significantly reduces the risk of respiratory failure (16%). Concerning OHS (obesity hypoventilation syndrome), hypercapnia is the main determinant and is characterized by an increased respiratory workload. This results in a decrease in pulmonary compliance (lamellar atelectasis), a decrease in thoracic wall compliance (diaphragmatic elevation), and a limitation of respiratory flow (supine position). Leptin resistance, decreased production of leptin and bicarbonate retention, and resetting of chemosensitivity during nocturnal periodic breathing contribute to the reduced ventilatory response to hypercapnia and/or hypoxia (typical in obesity). Patients with hypercapnia from diagnosed or suspected OHS are at higher risk for postoperative respiratory failure, postoperative heart failure, prolonged intubation, postoperative intensive care unit (ICU) transfer, and longer ICU and hospital stay when compared to patients with OSA. Another study [3] aimed to demonstrate to what extent obese patients diagnosed with OSA and undergoing bariatric surgery are at an increased risk of hypoxemia in the first 24 h after laparoscopy.

The anesthetic protocol is standardized using propofol for induction and remifentanil for maintenance. Postoperative pain management is based on the controlled-programmed administration of morphine. In a study of 40 patients, of which 31 were diagnosed with OSA, with a body mass index (BMI) between 35 and 70 kg/m^2^ who were scheduled for bariatric surgery, polysomnography was performed for four weeks prior to the intervention. Preoperatively, subjects with OSA clearly recorded a higher desaturation index during PSG compared to those without OSA. Pulse oximetry monitoring in the first 24 h after surgery did not show significant differences in average saturation with or without the use of oxygen between subjects with or without sleep apnea. The number of desaturations did not vary significantly between the two groups [20]. Ventilation devices corrected desaturations in patients with sleep apnea, resulting in both categories of patients having an equal number of events (those occurring due to obesity).

Another study, which was conducted over a period of 3 years and involved 410 patients scheduled for bariatric surgery and previously examined through nocturnal pulse oximetry, addressed and discussed aspects related to the presence or absence of OSA and the use or the nonuse of continuous positive airway pressure [19]. Mortality and perioperative complications were the main outcomes measured. Over 2/3 (70%) of the tested patients had been diagnosed with sleep apnea, and 40% of the patients included in the study used CPAP. Regarding the duration of hospitalization and the rate of respiratory complications, there were no significant differences between those who used CPAP and those who did not. In practice, bariatric patients who are appropriately screened and treated according to the guidelines do not have an increased risk of respiratory complications compared to those who do not have OSA [3].

Patients diagnosed and suspected of having OSA should be managed with a systematic algorithm in order to improve their outcomes [6]. Sleep apnea may be under-recognized in surgical populations [21], and before any surgical intervention, it is necessary to perform a polysomnography if the patient is at risk of OSA. If the diagnosis is confirmed, treatment with positive pressure ventilation must be administered prior to surgical intervention. Afterwards, adherence to treatment must be checked as well as the effectiveness of the therapy. The suitability of ambulatory surgery in patients with OSA remains controversial, and the evidence regarding the safety of ambulatory surgery for patients with OSA is limited [6].

### 3.7. CPAP vs. BPAP

In the treatment of sleep-related upper airway obstruction, the treatment goals are to reduce the upper airway resistance during sleep, decrease the breathing workload, and increase nocturnal ventilation. CPAP improves nocturnal and diurnal hypoventilation and nocturnal oxygen saturation in obstructive sleep apnea (OSA). On the other hand, BPAP corrects respiratory events using a lower expiratory pressure (EPAP) and adjusts inspiratory pressure (IPAP) to correct hypoxemia. BPAP is recommended when high pressures are required that are difficult for the patient to tolerate when hypoxemia and hypercapnia persist despite APAP (automatic positive airway pressure)/CPAP therapy or in patients with obesity hypoventilation syndrome without sleep apnea. It is typically considered the second option after a failed CPAP trial. In OHS, noninvasive ventilation (NIV) improves arterial blood gas levels, respiratory symptoms, respiratory function, sleep quality, and carbon dioxide sensitivity. One disadvantage of BPAP is that, unlike CPAP, it has a higher cost associated with it.

CPAP treatment is effective for patients with sleep apnea syndrome with obesity without hyperventilation during REM sleep and moderate hypercapnia in general. Predictive factors for CPAP failure (compared to BPAP) include maintaining oxygen saturation below 80% for over 10 min in the absence of respiratory events, marked hypercapnia during REM sleep episodes, and morning hypercapnia compared to the rest of the day [4].

Adherence to CPAP is the key to a successful therapy, and it depends on several factors, with psychological factors being paramount. The patient’s perception of the disease, self-efficacy (confidence in the patient’s ability to make a change in their life), poor risk perception, and lack of perception of therapy benefits are impactful factors. Limited socioeconomic status and lack of health insurance coverage in certain countries are also important elements. Issues in the oropharyngeal area, lack of accommodation with the device or the mask (too big/too small), claustrophobia, the impact on intimate life, and shift work are other causes of nonadherence. Initiating therapy through home titration leads to higher adherence compared to laboratory testing [4].

In this study, we summarize the main aspects and considerations for the perioperative management of OSA, a condition that has rapidly become a public health concern. Critical determinants of perioperative risk include OSA-related changes in the upper airway anatomy with augmented collapsibility, diminished capability of upper airway dilator muscles to respond to airway obstruction, disparities in hypoxemia and hypercarbia arousal thresholds, and instability of ventilatory control. Perioperative screening to identify patients at risk has been implemented in many institutions, and caregivers who participate in the perioperative preparation of these patients should anticipate difficult airway management in OSA and also be prepared for airway complications.

Anesthetic and sedative drug agents usually worsen upper airway collapsibility and depress central respiratory activity, while the risk for postoperative respiratory compromise is further increased with the utilization of neuromuscular blockade. Consistently, opioid analgesia has been proven complex in OSA as patients are particularly prone to opioid-induced respiratory depression. Moreover, basic features of OSA, including intermittent hypoxemia and repetitive sleep fragmentation, gradually precipitate higher sensitivity to opioid analgesic potency along with an increased perception of pain. Hence, regional anesthesia by blockade of neural pathways directly at the site of surgical trauma and multimodal analgesia by facilitating additive and synergistic analgesic effects are both strongly supported in the literature as interventions that may reduce perioperative complication risk. Healthcare institutions are increasingly allocating resources, including for postoperative enhanced monitoring, in an effort to increase patient safety. The implementation of evidence-based perioperative management strategies is, however, burdened by the rising prevalence of OSA, large heterogeneity in the disease severity, and lack of evidence on the efficacy of costly perioperative measures. Screening and monitoring algorithms, as well as reliable risk predictors, are urgently needed to identify OSA patients that are truly in need of extended postoperative surveillance and care. The perioperative community is therefore challenged to develop feasible pathways and measures that can confer increased patient safety and prevent complications in patients with OSA [4].

In the case of severely obese patients with sleep apnea, positive pressure ventilation can lead to weight loss. Consequently, after a certain period of time, during repeated polysomnography, a de-escalation to moderate apnea is observed simply by removing the primary risk factor of obesity. However, it requires time, treatment adherence, and compliance from the patient. In applicable cases, interventions in the field of otorhinolaryngology (ENT) have curative potential by addressing the causal factor. Thus, after surgery, the apnea disappears, but the risks associated with the intervention must be stated, made aware of, or assumed. Nevertheless, as mentioned before, the use of CPAP is necessary before the actual intervention.

The literature debating the benefits of using CPAP in the preoperative and postoperative periods in patients with SASO is insufficient. Therefore, the perioperative guidelines are based on combinations of randomized trials, observational studies, and case studies.

In a review summarizing 21 randomized and observational trials, the role of CPAP in postoperative physiology and the occurrence of complications was extensively highlighted. The authors emphasized that the use of CPAP after surgery improves oxygenation and reduces the need for reintubation and mechanical ventilation. Additionally, respiratory events (apnea and hypopnea), as well as postoperative hypoxemia, are reduced. However, low adherence to CPAP is a negative predictive factor when it comes to cardiovascular impact [10].

The perioperative care of obstructive sleep apnea (OSA) patients is currently receiving much more attention due to the increased risk of complications. Postoperative changes in sleep architecture occur, and this may have pathophysiological implications for OSA patients. The upper airway muscle activity decreases during rapid eye movement sleep (REMS), and severe OSA patients exhibit exaggerated chemoreceptor-driven ventilation during nonrapid eye movement sleep (NREMS), which leads to central and obstructive apnea [20]. Anesthetic, sedative, and analgesic drugs have been shown in animals and humans to selectively impair upper airway muscle activity. In patients with an already compromised upper airway, these drugs may further jeopardize upper airway patency, especially during sleep.

Patients with obstructive sleep apnea syndrome present a much higher risk for surgery also because of the use of the aforementioned drugs in the perioperative period. In a report on 16 cases, patients diagnosed with sleep apnea who were undergoing various types of surgical procedures were evaluated from the perspective of preoperative CPAP treatment [22]. The anesthesia was carried on with the usual type of drugs for each type of surgery without restricting the sedation and postoperative opioid analgesia use. One of the patients who was not treated for sleep apnea suffered a respiratory arrest in the ward and died, but another patient with serious complications had an uneventful recovery after treatment for OSAS with N-CPAP was instituted. None of the patients using N-CPAP before surgery had complications. They continued using therapy after surgery as soon as extubated for 24–48 h and thereafter for all sleep periods, highlighting the benefits of using N-CPAP preoperatively and immediately after extubation in patients with OSAS [23].

During the anesthesia and sedation, patients with upper airway obstruction during sleep are at risk of hypoxic and hypercapnic episodes and are especially vulnerable. Of course, the abnormal anatomy is compounded by drug-related respiratory depression. Respiratory supportive techniques started at home should be continued during hospitalization, both in the preoperative and postoperative periods [24]. Patients with OSA should be examined for possible anatomic abnormalities of the upper airway, which may complicate laryngoscopy and/or intubation. Careful preoxygenation is needed and opioids should be used sparingly, while isoflurane should be avoided and muscle relaxants should be calculated for an ideal body wight [25]. Some studies have advanced the idea that in order to avoid limitation of the functional residual capacity of the lungs (FRC) by an elevated diaphragm, obese patients should be extubated in sitting or lateral positions; however, in selected cases, prolonged intubation or ventilation is recommended. Although regional anesthesia is usually safe for these patients, opioids should be used carefully when sedation is required but ketamine or dexmedetomidine may be used [26].

Regarding anesthetic risk, a case report from 2000 discussed the particularity of a 39-year-old patient with a BMI of 34.22 kg/m^2^ and known to have sleep apnea who was undergoing CPAP treatment and had been scheduled for an elective procedure in the plastic surgery department. After anesthesia induction and direct laryngoscopy, adequate airflow could not be ensured should the patient become hypoxic, hypercapnic, bradycardic, and hypotensive. The team used a laryngeal mask and stabilized the patient who recovered with no immediate neurological sequelae upon awakening. The following fiberoptic intubation of the awake patient, still following the tendency of upper airway obstruction, confirmed the challenging anatomical structures, but the subsequent general anesthesia was uneventful. The patient then received CPAP and was monitored in the intensive care unit during the first postoperative night and recovered without sequelae. Here, the medical team was advised to opt for regional anesthesia, only confirming that the sleep apnea syndrome poses an important challenge for the anesthesiologist, especially given patients with sleep apnea are suspected to have associated pathologies, especially cardiovascular ones. It is important to avoid sedatives and initiate CPAP before surgery, and postoperative opioid analgesics should be administered under strict supervision [27].

In the field of ENT, uvulopalatopharyngoplasty is a therapeutic method for sleep apnea as it removes the causal factor.

A study conducted over the course of one year aimed to highlight the perioperative risks and complications in 32 patients undergoing general anesthesia for such a procedure with or without septoplasty. Criteria such as difficult intubation, reintubation, postoperative pulmonary edema, postoperative desaturations, and the need for positive pressure ventilation were evaluated. None of the subjects required reintubation or prolonged intubation, and none developed postoperative pulmonary edema. Cases with difficult intubation did not develop any other complications. Thus, the intervention proves to be curative and devoid of major risks [28].

Another study [29] added an extra level of safety to the method. It demonstrated a low rate of postoperative complications, eliminating the need for overnight hospital observation. The study included 117 patients who underwent the procedure over a 5-year span. Respiratory events occurred in 2–11% of cases, but airway obstruction and postoperative pulmonary edema (POPE) occurred immediately after the procedure. Desaturations occurred at various intervals but at the same parameters as preoperative recordings, indicating that the intervention did not worsen the condition. Hemorrhages occurred in a biphasic manner, either immediately after the procedure or a few days later. Hypertension was identified and treated immediately following the intervention. Arrhythmia was identified in less than 1% of cases. Therefore, it can be concluded that these complications occur within 1–2 h after the procedure [29].

A meta-analysis conducted in 2022 compared the benefits of CPAP versus mandibular advancement devices in the treatment of moderate and severe obstructive sleep apnea (OSA). The AHI (apnea–hypopnea index), minimum nocturnal saturation, and daytime sleepiness (assessed using the Epworth sleepiness scale questionnaire before and after treatment) were evaluated. The patients underwent treatment for various periods of time, and the study’s final conclusion was that CPAP therapy has clear advantages in terms of AHI improvement and minimum nocturnal saturation but both types of treatment have the same impact on the sleepiness scale [21].

Screening patients for sleep apnea must be performed before any surgical intervention for high-risk groups. Therefore, a study conducted in 2018 aimed to evaluate the risk of postoperative complications after oromaxillofacial surgeries. Sixty-nine patients known to have OSA were compared to a control group without OSA, with both receiving the same treatment. Almost 1/3 of patients with OSA (29%) developed respiratory complications, starting from the intensive care unit, in contrast to the control group (9%) [20].

Patients with OSA may be at increased perioperative risk, in part due to the effects of sedatives and anesthetics on the upper airway tone and respiratory drive as a growing amount of data suggests that OSA patients have increased odds for adverse postoperative outcomes, including intensive care unit transfer, respiratory failure, arrhythmias, and cardiac ischemia [30]. Patients with OSA may require additional monitoring, unplanned escalations in care, and prolonged hospitalization as they have increased risk of respiratory, cardiac, and infectious complications following surgical procedures [1].

### 3.8. Having a Diagnosis of Sleep Apnea While Surviving the COVID-19 Pandemic

Since February–March 2020, the issue of the COVID-19 pandemic has occupied the agenda of the whole world, unintentionally disregarding the chronic and somewhat managed known diseases with a functional treatment prescribed.

Although the rationale for lockdown was well sustained by the strong, growing (but sometimes exaggerated) epidemiological arguments, exploring the other unwanted consequences of the contemporary COVID-19 pandemic had an impact on sleep apnea syndrome as well. There were numerous problems generated by the SARS CoV-2 viral contention as the major objective of any quarantine is to reduce contagion or spread of disease [31]. However, the restrictive measures were “the unnamed enemy” for other conditions, especially for elective medical appointments. Sleep laboratory investigations were largely suspended as were the diagnostic steps for sleep apnea and the initiation of therapy for sleep apnea patients. Sleep management was also affected during the COVID-19 pandemic in patients, their families, and the isolated population, and especially for patients who already suffered from sleep disorders. Obstructive sleep apnea was found to be a frequent baseline characteristic of COVID-19 patients [32], and we are obligated to raise the question of whether these patients would have had a different outcome after COVID-19 disease if their sleep apnea was correctly managed. Would they have suffered less or even, in some unfortunate cases, be alive today had their CPAP or BPAP home ventilation treatment been instituted? More data and analysis are needed on the matter, but we believe that this idea is worth pursuing.

The pandemic can be seen as a total eclipse of all other diseases for some cases. According to the WHO, more than half of the countries have partially or completely discontinued routine checks for hypertension treatment, while 49% have done so for treatment for diabetes and diabetes-related complications, 42% for cancer treatment, and 31% for cardiovascular emergencies [31].

## 4. Discussion

The debate over obstructive sleep apnea syndrome and obesity hypoventilation is always complexity and largely refers to comorbidities. In terms of multidisciplinarity, patients are usually referred to sleep laboratories by their family doctor, cardiologist, ENT specialist, or ophthalmologist.

This paper, in an attempt to merely scratch the surface of an interdisciplinary approach regarding sleep apnea syndrome management, aimed to also initiate a debate: Can the advantages of surgical therapy outweigh the disadvantages in the case of OSAS?

Prior to any elective surgical intervention, individuals with risk factors for OSAS should be referred to a sleep specialist for a sleep test. If the diagnosis is confirmed, it is mandatory to undergo positive pressure ventilation for 4–6 weeks before surgery. Surgical treatment will only be performed if there is evidence of adherence to treatment, tolerance of the ventilation device, and effectiveness of treatment. In practice, by reviewing the information on the compliance card, we can determine whether the patient is suitable for anesthesia and would benefit from intervention. The prevalence of OSA implies a significant risk for perioperative settings as patients are vulnerable to cardiopulmonary complications, critical care requirements, and unexpected death [33].

In addition, the advantages of positive pressure ventilation and noninvasive ventilation are extensively discussed. Adherence to CPAP is the key to therapeutic success and depends on several factors, with psychological factors being paramount. The patient’s perception of the disease, self-efficacy (confidence in the patient’s ability to make a change in their life), poor risk perception, and lack of perception of therapy benefits are impactful factors. Limited socioeconomic status and lack of health insurance coverage in certain countries are also important elements. A low tolerance to CPAP can also be caused by issues in the oropharyngeal area. Replacing a nasal mask with an oronasal mask can be beneficial [16]. Initiating therapy through home titration leads to higher adherence compared to laboratory testing.

Patients with OSA may be at increased perioperative risk, also due to the effects of sedatives and anesthetics on the upper airway tone and respiratory drive [32]. It increases the risk of respiratory, cardiac, and infectious complications following surgical procedures. Patients with OSA may require additional monitoring, unplanned escalations in care, and prolonged hospitalization [1].

This paper summarizes the main aspects and considerations for the perioperative management of OSA, which has become a public health concern. Critical determinants of perioperative risk include OSA-related changes in the upper airway anatomy with augmented collapsibility, dismissed capability of upper airway dilator muscles to respond to airway obstruction, disparities in hypoxemia and hypercarbia arousal thresholds, and instability of ventilatory control [34]. Perioperative screening to identify patients at risk has been implemented in many institutions. Perioperative caregivers should anticipate difficult airway management in OSA and be prepared for airway complications [35].

## 5. Conclusions

As this paper is a debate, the conclusions remain as such. Generalization is not possible; instead, each case has to be individualized while considering all risks. The approach to patients with sleep apnea or obesity hypoventilation is undoubtedly multidisciplinary. There is no ideal method of treatment; each therapeutic opportunity has its advantages and disadvantages and must be personalized and adapted for each patient. Only then can the optimal therapeutic approach be selected depending on the type and severity of apnea as well as the presence of obesity hypoventilation syndrome. The treating physician will take into account comorbidities, risk factors, and the underlying cause of sleep apnea.

As demonstrated by the results of the studies mentioned above, positive pressure ventilation represents the therapeutic gold standard for sleep apnea. Depending on the type and severity of the apnea case, a specific type of device (CPAP/BPAP) is selected. As extensively mentioned in the present paper, adherence to ventilation is the key to success, and it depends on many factors. On the other hand, the disadvantages of CPAP/BPAP are also discussed, such as the psychological impact, nightly use, and costs.

Surgical treatment, whether aimed at addressing the causal factor (ENT or maxillofacial interventions) or curative treatment of obesity, has its own risks and benefits, which need to be carefully analyzed. This paper emphasizes the anesthetic risks but also highlights the fact that patients are screened for OSA prior to any surgery. Thus, the topic of ventilation is always brought up.

Multidisciplinary management, from identifying patients with risk factors to approaching them for establishing therapeutic strategies, is mandatory. This study aims to become the foundation for raising awareness of the need to develop a clinical practice guideline for a multidisciplinary approach to patients with sleep apnea.

## Data Availability

The data published in this research are available on request from the first author and corresponding author. All information provided in this review is documented by relevant references.

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
