# Peer review of "A Debate on Surgical and Nonsurgical Approaches for Obstructive Sleep Apnea: A Comprehensive Review"

_jpm, 2023, doi:10.3390/jpm13091288_

Round 1

Reviewer 1 Report

the OHS should not in this manuscript, because of the title. the article at one point turned to comparing patients with OSA and OHS.
Bariatric surgery, anesthetic risks, postoperative complications were also discussed due to OHS.
Author should prefer OSA or OSAS.

Author Response

Response to Reviewer 1 Comments, article jpm-2571929 (Zabara-Antal et al.):

Dear Reviewer, thank you for your valuable comments. We respond punctually below.

The OHS should not in this manuscript, because of the title. the article at one point turned to comparing patients with OSA and OHS. Bariatric surgery, anesthetic risks, postoperative complications were also discussed due to OHS. Author should prefer OSA or OSAS.

It is correct, we entitled the article in reference to a debate on surgical/nonsurgical approach to obstructive sleep apnea, but we believe that not discussing obesity hypoventilation syndrome (OHS), the common risk factor for this sleep disorder, the most common etiological factor for the disease and the one syndrome that also maintains the disease until treated, would make the manuscript incomplete. Aside from a few particular situations, obesity, and all the pathology related to it, leads the clearest path to OSAS. The proposed and discussed surgical methods refer to obesity in relation to its association with sleep apnea. Otherwise, a debate on surgical or non-surgical interventions would have no sense. The majority of patients with obstructive sleep apnea are obese, many times they need bariatric surgery, and hence, our intention of not leaving OHS aside from the discussions of this article. The presented surgical curative methods do not address obese patients who do not have sleep apnea, as the authors have no competency of discussing surgical treatments for other metabolic diseases or state other benefits of bariatric surgery outside the indication for obesity treatment to reduce or permanently cure sleep apnea (which, in this case, is the most life-threatening disease). All mentions related to the management of obese patients have been made in association with the presence or absence of this sleep disorder, in order to determine the significance of sleep apnea in the patient’s progress. As stated in a paper from 2017 (https://doi.org/10.18632%2Foncotarget.21450) “[…] More importantly, the severity of OSA and the impairment of respiratory system mechanics were closely associated with the occurrence of OHS. […] Obesity has an influence on both hypoventilation and apnea. OHS and OSA always coexist. […]” Also, there is plenty of data suggesting that surgical treatment of the overlapping OHS and OSA must be considered as a curative intervention for both diseases (https://www.archbronconeumol.org/en-the-overlap-obesity-hypoventilation-syndrome-obstructive-articulo-S0300289621002301, https://doi.org/10.1097/ALN.0b013e31825add60)

We have modified, in the text, as much as possible, OHS with OSA/OSAS  (of course, only where this was appropriate). We hope that our explanations sustain the fact that healthcare professionals should consider both diseases as a “comorbidity burden” and address them bilaterally.

Thank you again for your valuable comment, we adapted the text.

Reviewer 2 Report

The authors have made a narrative review of the surgical and non-surgical treatment for patients with OSA. 

I have some comments regarding the manuscript.

Line 27, OHS has no name before, 

line 59 there's a DOI instead of a reference number

line 82: PSG is not required for a diagnostic of OSA, a sleep study is, and home sleep studies, as you say later, are perfectly valid for patients with high pretest probability and no comorbidities.

line 109: another reference not well documented 

line 110-117: probably you should specify what you consider individuals with risk factors for OSA, who refers the patient for a sleep study, the waiting list to do the study, and how you manage everything so the patients have at least one month of CPAP before any elective surgery, or it's just for bariatric surgery? Because what you say happens may look like an optimistic and unrealistic thing in many places due to the waiting list for sleep studies and specifically for PSG. 

line 137 if you are going to say something like that that it's not true for high probability patients put a reference otherwise delete it

line 180 it's Pringle and Croft

line 185-186: put the reference

line 191: actually tongue base is in the oropharynx 

line 198: CO2 laser is not a valid treatment according to the AASM, I believe that it's the only treatment that they say should be avoided due to the poor results. 

line 209: MAD can actually reverse palatal collapse too, they have proven that they achieve blood pressure reductions and increase QoL as CPAP. DISE it's not necessary for titration but can be used. What it's absolutely necessary it's a post-titration sleep study to know if it's useful

line 214 tongue advancements/stabilization has little room in OSA, and very few publications. Usually, people don't use them

line 2017: which criteria for positional have you used: reference

line 241: again PSG it's not necessary, a sleep study is

line 258: ICU: explain letters

line 271: here you explain it but later and shouldn't

line 281-84: explain that they don't change because the CPAP avoids desaturations 

line 306: sleep-related upper airway obstruction (OHS) letters don't match what it's written

line 452: the reference it's too old there are new articles that may not agree with this statement

The side effects that you mention of CPAP are mostly psychological side effects, but there are more problems, some patients don't tolerate them due to other problems like epiglottic obstruction, no resolution of the obstruction, etc. There are references for this too.

I believe that you should explain better what type of surgery you are talking, about because if it's for every type of elective surgery like cholecystectomy, trauma... the problem of the sleep studies and waiting list is an important one.  

Author Response

Response to Reviewer 2 Comments, article jpm-2571929 (Zabara-Antal et al.):

Dear Reviewer, thank you for your valuable comments. We respond punctually below.

  • Line 27, OHS has no name before

Authors made some changes, so the term is not mentioned on this line anymore, it first appears on line 251 (and is explained there).

  • line 59 there's a DOI instead of a reference number

We have introduced the reference number missing (6).

  • line 82: PSG is not required for a diagnostic of OSA, a sleep study is, and home sleep studies, as you say later, are perfectly valid for patients with high pretest probability and no comorbidities.

We’ve changed the PSG with the sleep test/sleep study in the text.

  • line 109: another reference not well documented

Reference number is 11( we have modified in text)

  • line 110-117: probably you should specify what you consider individuals with risk factors for OSA, who refers the patient for a sleep study, the waiting list to do the study, and how you manage everything so the patients have at least one month of CPAP before any elective surgery, or it's just for bariatric surgery? Because what you say happens may look like an optimistic and unrealistic thing in many places due to the waiting list for sleep studies and specifically for PSG.

A new paragraph bas been added to the text. It is mentioned which patients are considered to have risk factors for sleep apnea (obese, ENT pathology, specific day and night symptoms, who refers them to sleep test(the anesthesiologist during the pre-anesthetic consultation). It outlines the management of these patients. Sleep medicine and sleep laboratories development is leading to increased accessibility. Patients are now able to benefit from prompt investigations. The accessibility and adherence to treatment are continuously increasing.

  • line 137 if you are going to say something like that that it's not true for high probability patients put a reference otherwise delete it

We deleted the phrase intended for high- risk probability patients, as the reviewer recommended.

  • line 180 it's Pringle and Croft

We’ve mentioned- ‘according to Pringle and Croft’.

  • line 185-186: put the reference

The reference number is now introduced (17).

  • line 191: actually tongue base is in the oropharynx

We have corrected the term as the reviewer suggested (the used term now is ‘oropharynx’).

  • line 198: CO2 laser is not a valid treatment according to the AASM, I believe that it's the only treatment that they say should be avoided due to the poor results.

The laser treatment phrase was deleted.

  • line 209: MAD can actually reverse palatal collapse too, they have proven that they achieve blood pressure reductions and increase QoL as CPAP. DISE it's not necessary for titration but can be used. What it's absolutely necessary it's a post-titration sleep study to know if it's useful

We’ve mentioned now that MAD can reverse palatal collapse too. Also, it is specified that DISE is less commonly used. Also, the need for the sleep test is explained.

  • line 214 tongue advancements/stabilization has little room in OSA, and very few publications. Usually, people don't use them

As the reviewer recommended, we’ve explained that tongue advancements are not so often used.

  • line 2017: which criteria for positional have you used: reference

Now the quoted paragraph no longer refers to positional sleep apnea.

  • line 241: again PSG it's not necessary, a sleep study is

We have changed the term (sleep test/sleep study).

  • line 258: ICU: explain letters

The explanation is now in the text.

  • line 271: here you explain it but later and shouldn't

We have corrected in the text. The explanation is on the 258 line

  • line 281-84: explain that they don't change because the CPAP avoids desaturations

Authors introduced this phrase: ‘Ventilation devices have corrected desaturations in patients with sleep apnea, resulting in both categories of patients having an equal number of events (those occurring due to obesity).

  • line 306: sleep-related upper airway obstruction (OHS) letters don't match what it's written

We have deleted the abbreviation in the text.

  • line 452: the reference it's too old there are new articles that may not agree with this statement

We consulted newer references and updated this item.

  • The side effects that you mention of CPAP are mostly psychological side effects, but there are more problems, some patients don't tolerate them due to other problems like epiglottic obstruction, no resolution of the obstruction, etc. There are references for this too. I believe that you should explain better what type of surgery you are talking, about because if it's for every type of elective surgery like cholecystectomy, trauma... the problem of the sleep studies and waiting list is an important one.

We’ve expanded the paragraph about CPAP adherence, as the reviewer recommended. There are new references here (for instance, line 519). Also, we have explained that the process is the same, for bariatric surgeries or other elective surgical intervention. The text explains how, in recent times, the field of sleep medicine and sleep laboratories have experienced significant development. So, the waiting list is not such a problem nowadays (line 121).

We thank you oce again for your valuable inputs and hope that the corrections we made are in order.

Round 2

Reviewer 1 Report

thank you

Author Response

 Article jpm-2571929 (Zabara-Antal et al.):

Dear Reviewer, thank you for your valuable time. We are glad that we could follow your recommendation. 

Reviewer 2 Report

The authors have made some changes but I still see some mistakes, I will refer to the lines of the new manuscript with the red corrections.

Line 186 delete according to Pringle and Soft as they didn't publish about the position

Line 196, it's Pingle and Croft not soft. 

line 274 the low tolerance to CPAP it's not only due to ENT problems, I see many patients who just have claustrophobic problems or other types. 

lines 478-82: although there's that study that says that laser can help, as it is not recommended, I believe that it would be best if you could delete that sentence. There are new  references that can be used instead that tell what you want to say, but with no laser

Author Response

Article jpm-2571929 (Zabara-Antal et al.):

Dear Reviewer, thank you for your valuable comments. We respond punctually below.

  • Line 186, delete according to Pringle and Soft, as they didn’t publish about the position

Authors deleted, as you recommended.

  • Line 196, it’s Pringle and Croft not soft

We’ve corrected that mistake.

  • line 274: the low tolerance to CPAP it’s not only due to ENT problems, I see many patients who have claustrophobic problems or another types.

Issues in the oropharyngeal area, lack of accommodation with the device or the mask (too big/to small), claustrophobia, the impact on intimate life, work in shifts, are other causes of non-adherence mentioned now in the text, completing the existing information. You can find a phrase on line 320.

  • Lines 478-82: although there’s that study that says that laser can help, as it is not recommended, I believe that it would be best if you could delete this sentence. There are new references that can be used instead that tell what you want to say, buy no laser.

Authors deleted the entire phrase.